# Potential Biomarkers in Systemic Sclerosis: A Literature Review and Update

**DOI:** 10.3390/jcm9113388

**Published:** 2020-10-22

**Authors:** Akira Utsunomiya, Noritaka Oyama, Minoru Hasegawa

**Affiliations:** Department of Dermatology, Divison of Medicine, Faculty of Medical Sciences, University of Fukui, 23-3, Matsuokashimoaizuki, Eiheiji-cho, Yoshida-gun, Fukui 910-1193, Japan; utsunomi@u-fukui.ac.jp (A.U.); norider@u-fukui.ac.jp (N.O.)

**Keywords:** systemic sclerosis, biomarker, fibrosis, vascular injury, autoantibody

## Abstract

Systemic sclerosis (SSc) is a chronic autoimmune disease characterized by dysregulation of the immune system, vascular damage, and fibrosis of the skin and internal organs. Patients with SSc show a heterogeneous phenotype and a range of clinical courses. Therefore, biomarkers that are helpful for precise diagnosis, prediction of clinical course, and evaluation of the therapeutic responsiveness of disease are required in clinical practice. SSc-specific autoantibodies are currently used for diagnosis and prediction of clinical features, as other biomarkers have not yet been fully vetted. Krebs von den Lungen-6 (KL-6), surfactant protein-D (SP-D), and CCL18 have been considered as serum biomarkers of SSc-related interstitial lung disease. Moreover, levels of circulating brain natriuretic peptide (BNP) and N-terminal pro-brain natriuretic peptide (NT-proBNP) can provide diagnostic information and indicate the severity of pulmonary arterial hypertension. Assessment of several serum/plasma cytokines, chemokines, growth factors, adhesion molecules, and other molecules may also reflect the activity or progression of fibrosis and vascular involvement in affected organs. Recently, microRNAs have also been implicated as possible circulating indicators of SSc. In this review, we focus on several potential SSc biomarkers and discuss their clinical utility.

## 1. Introduction

Systemic sclerosis (SSc) is a multifaceted chronic autoimmune disease with characteristic vascular damage and fibrosis of the skin and internal organs [1,2]. Diagnosis of SSc is mainly based on the clinical course and features in addition to laboratory findings including autoantibody profiles. SSc patients can be classified clinically into two groups, limited cutaneous systemic sclerosis (lcSSc) and diffuse cutaneous systemic sclerosis (dcSSc), according to the peak extent of skin involvement [3]. Skin thickening in lcSSc patients is limited to the face and the distal aspect from the elbows and knees, whereas that of dcSSc patients involves the trunk and proximal extremities in addition to distal areas. Patients with dcSSc generally show more severe and rapid progression of skin thickening and internal organ involvement including interstitial lung disease (ILD), cardiovascular lesions, and renal crisis. Vascular involvement, including digital ulcers and pulmonary arterial hypertension (PAH), can be detected in both lcSSc and dcSSc.

The clinical course of skin sclerosis and organ lesions varies in each case. Currently, SSc-related mortality is predominantly due to ILD and PAH [4]. Recently nintedanib, a small molecule tyrosine kinase inhibitor, has been approved for the progressive SSc-associated ILD (SSc-ILD) [5]. However, successful treatment of SSc remains challenging and immunosuppressive therapies, such as cyclophosphamide, methotrexate, mycophenolate acid mofetil, may not provide the prognostic improvement, particularly in dcSSc [6]. The potential difficulty on the therapeutic management includes the irreversible damage of fibrosis and vascular injury in the affected organs. Therefore, it is essential to distinguish clinically high risk patients in order to identify predictors of the evolution of the disease, for example to predict the onset of ILD in one patient and more importantly to predict the evolution of ILD. A comparative summary is provided in Table 1.

## 2. Objective and Methods

In this article, we aimed to literature review major candidate biomarkers in SSc according to their distinct biological actions. A PubMed search for articles published between January 1981 and August 2020 was conducted using the following keywords: “systemic sclerosis” and “biomarker”. From those references, a possible broad spectrum of biomarkers were selected subjectively. Some reference lists of identified articles were searched for further articles.

## 3. Autoantibody

Similarly to what is seen in other connective tissue diseases, most SSc patients are seropositive for antinuclear antibodies and have disease-specific autoantibodies, which can be detected prior to the development of clinical symptoms [7]. Disease-specific autoantibody profiles support not only conclusive and phenotypic diagnoses, but can also be associated with clinical manifestations and disease progression of SSc. In fact, autoantibodies such as anticentromere antibody, anti-topoisomerase I antibody, and anti-RNA polymerase III antibody are specific and included in 2013 ACR/EULAR criteria [8].

### 3.1. Anti-Topoisomerase I Antibody

The presence of anti-topoisomerase I Ab is generally associated with diffuse cutaneous involvement, severe fibrotic changes of internal organs, vascular injury such as digital ulcers, and to a lesser extent with PAH [9,10]. SSc patients with this antibody frequently develop SSc-ILD and tend to exhibit a decline in forced vital capacity (FVC) within the initial 3 years of clinical course [11]. A recent large population study assessing the combination of antibody type and disease subset (dcSSc and lcSSc) in relation to prognosis demonstrated the lowest 20-year survival rate in dcSSc with anti-topoisomerase I antibody [12]. However, lcSSc with this antibody was indicative of the second highest survival rate next to lcSSc with anticentromere antibody among several combination groups.

### 3.2. Anti-RNA Polymerase III Antibody

Anti-RNA polymerase III antibody correlates with increased risk of dcSSc, renal crisis, joint contractures, and is often associated with the complication of malignancies [13,14]. Moreover, presence of this antibody is associated with rapidly increasing skin thickening and a shorter interval between the first appearance of any SSc-related symptoms and peak skin thickness [15]. The overall prevalence of renal crisis during the course of disease ranges from 14% to 51%. A large cohort study suggested the potential association between anti-RNA polymerase III antibody and gastric antral vascular ectasia, so called “watermelon stomach” [16]. Additionally, patients with this antibody exhibited malignancy rates of 31.8–17.7% compared to 5.8–2.4% in those with other antibodies or in healthy controls [17,18,19]. Anti-RNA polymerase III antibody are associated with malignancies, but only with malignancies synchronous to SSc onset, that is currently their main key role as biomarkers. If such positivity is found, clinicians should seek for the presence of an occult cancer (if not already manifested) and monitor for the onset of renal crisis.

### 3.3. Anticentromere Antibody

Anticentromere antibody is detected in about 70% of SSc but may also be found in patients with other collagen diseases and primary biliary cholangitis [20,21]. While for SSc this antibody is associated with modest fibrosis of the skin (lcSSc) and internal organs [22,23], it is also associated with slowly progressive vascular involvement such as digital ulcers and PAH [24,25].

### 3.4. Other Autoantibodies

SSc patients with anti-U3 RNP antibody generally progress to dcSSc and can develop skeletal myopathy, cardiac involvement, and PAH [12,26]. Anti-Th/To antibody is commonly found in lcSSc, particularly with PAH and ILD [27,28]. Anti-U1 RNP antibody is not specific for SSc and SSc patients with this antibody often develop systemic lupus erythematosus and/or myositis [29,30].

### 3.5. Anti-Endothelial Cell Antibodies

Anti-endothelial cell antibodies are found in a substantial number of SSc patients (22–86%) [31]. These antibodies have been shown to target various vascular antigens, such as ICAM-1 [32], lamin A/C, tubulin β-chain, and vinculin [33], which are responsible for endothelial cell activation via increased oxidative stress and proinflammatory responses. Some authors found anti-endothelial cell antibodies more frequently in SSc patients with ILD than in those without ILD [34].

## 4. Growth Factors

### 4.1. Transforming Growth Factor (TGF)-β

The characteristic hallmark of SSc is excessive accumulation of extracellular matrices composed of collagen, elastin, glycosaminoglycan, and fibronectin in tissues. TGF-β is believed to be a key mediator of SSc pathogenesis, and especially of disease-associated fibrosis [35,36]. The inactive precursor of TGF-β is secreted by macrophages and other cells including platelets, leukocytes, and fibroblasts and is transformed into the biologically active form downstream of integrin signaling. Following receptor binding, TGF-β signaling results in the recruitment of immune cells including macrophages, which in turn produce more TGF-β [37]. Increased TGF-β results in the differentiation of fibroblasts and other precursors into myofibroblasts, inducing further downstream activation to stimulate extracellular matrix production and inhibit metalloproteinase synthesis [38]. Moreover, fibroblasts and myofibroblasts stimulated by TGF-β continue to produce TGF-β in an autocrine manner.

Analysis by DNA microarray revealed that several TGF-β-dependent genes are overexpressed in skin lesions of SSc patients [39]. In addition, TGF-β1 and TGF-β2 mRNAs were found to be elevated in skin biopsy specimens of dcSSc patients [40]. These expression levels differed significantly between lesional and nonlesional samples as well as between SSc and healthy control samples [40]. In addition, increased expression of TGF-β-regulated genes has been confirmed in lung tissues of SSc patients with progressive ILD [41].

It should be noted that the utility of circulating TGF-β as a biomarker is controversial. One study found elevated plasma TGF-β1 in only 6 of 39 patients with SSc, and not in any patients with primary Raynaud’s disease or in healthy controls [42]. Another study reported lower serum levels of active TGF-β1 in dcSSc patients than in lcSSc patients or controls. Moreover, a negative correlation was found between skin score in dcSSc and TGF-β1 levels in serum of dcSSc patients and a positive correlation with disease duration [43]. On the other hand, a relatively recent study showed that serum TGF-β1 is increased in SSc patients and positively correlates with severe digital ulcers and extensive skin fibrosis. However, presence of this cytokine in serum did not correlate with lung involvement or disease severity [44]. Further studies will be needed to determine the significance of circulating TGF-β in SSc.

### 4.2. Platelet-Derived Growth Factor (PDGF)

PDGF also likely plays a critical role in the fibrotic process of SSc. Secreted by platelets, endothelial cells, macrophages, and fibroblasts, PDGFs are heterodimeric peptides that function as potent mitogens and chemo-attractants for mesenchymal cells. Patients with SSc exhibit increased expression of PDGF receptor subunits, PDGFRα and PDGFRβ, on cells in the skin and bronchoalveolar lavage (BAL) fluid, and have circulating autoantibodies that trigger PDGFR-mediated fibroblast activation and reactive oxygen species synthesis [45,46,47]. Furthermore, the nintedanib, a tyrosine kinase inhibitor which targets PDGF, fibroblast growth factor and vascular endothelial growth factor (VEGF), has recently been approved for the treatment of SSc-ILD [5].

### 4.3. Connective Tissue Growth Factor (CTGF)/CCN2

CTGF/CCN2 has been considered to be involved in TGF-β-induced fibrosis [48]. Normally expressed at low levels, CTGF is markedly increased in fibroblasts stimulated by TGF-β, endothelin 1, angiotensin II, and IL-4 [49]. Additionally, expression of CTGF is increased in the epidermis of SSc skin [50]. As a fibrogenic cascade downstream mediator of TGF-β1, CTGF enhances production of two predominant extracellular matrix components, collagen 1 and fibronectin, by fibroblasts [51]. A series of studies in mouse models suggest fibrosis occurs through two-steps in SSc with early fibrosis being induced by TGF-β1 and CTGF contributing to maintenance of the secondary phase [52].

Several CTGF gene polymorphisms are reportedly associated with SSc-ILD [53,54]. Additionally, gene expression profiling revealed CTGF is upregulated in skin from dcSSc patients compared with normal skin [55]. A significant increase in CTGF was seen in sera of patients with SSc, and the levels were found to correlate with the extent of skin sclerosis and severity of ILD [56]. A previous manuscript reported that CTGF overproduced by SSc microvascular endothelial cells induced an efficient fibroblast activation, resulting also in an increase in fibroblast migration [57]. CTGF effects were mediated by an up-regulation of the TGF-β system (ligand/receptors). Further, the CTGF/TGF-β-dependent increase in fibroblast mobilization was accounted for by augmentation of their mesenchymal style of migration. Taken together, these data indicate that endothelial cells of SSc patients may trigger fibrosis initiation by inducing a CTGF/TGF-β-dependent fibroblast mesenchymal-to-mesenchymal transition, that is, an increase of their mesenchymal properties.

### 4.4. Vascular Endothelial Growth Factor (VEGF)

Persistent upregulation of VEGF as a result of cytokine stimulation is believed to cause the disturbed vessel morphology seen in SSc skin [58]. Furthermore, impaired VEGF receptor signaling may also be contributory to vascular function seen in SSc may be a result of impaired VEGF receptor signaling [59]. VEGF concentrations as well as PDGF-AA, PDGF-BB, fibroblast growth factor (FGF)-2, and macrophage colony stimulating factor were all significantly increased in the lungs of potential candidates for lung transplantation with progressive fibrosing-ILD, including SSc-ILD, compared to donor lungs [60].

SSc patients display significantly increased serum VEGF levels with markedly high concentrations noted in patients with systemic organ involvement [59]. In addition, high serum VEGF in SSc was found to correlate with shorter disease duration [58], increased systolic pulmonary artery pressure [61], aggravated skin sclerosis, and reduced nailfold capillary density [62]. In contrast, a PRISMA-driven systemic review revealed that SSc patients with digital ulcers display lower serum VEGF levels, while significantly higher levels were detected at early clinical stages in SSc patients with digital ischemic manifestations [63]. Therefore, VEGF may be protective against peripheral ischemic manifestations. 

### 4.5. Growth Differentiation Factor 15 (GDF-15)

GDF-15 is a member of the TGF-β superfamily which exerts stimulatory and immunomodulatory actions on fibroblasts [64]. Serum levels of GDF-15 were found to be elevated in SSc-ILD, and to correlate negatively with respiratory function tests diffusion capacity for carbon monoxide (DLco) and FVC, as shown in a longitudinal, prospective study (*n* = 119) [65]. In addition, a correlation between serum GDF-15 levels and skin sclerosis, ILD, and PAH have been reported in SSc [66,67].

## 5. Cytokines

### 5.1. Interleukin-6

IL-6 is a multifunctional acute-phase inflammatory cytokine with an important role in the regulation of immune responses [68]. Produced by various cells, including leukocytes, fibroblasts and endothelial cells, IL-6 is involved in the pathology of various immune-mediated inflammatory diseases. While IL-6 plays a critical role in a wide variety of pathophysiologic processes, excessive production of this cytokine in SSc results in increased collagen production through fibroblast activation, myofibroblast differentiation, and inhibition of secretion of matrix metalloproteinases that carry collagenolytic activity responsible for tissue repair and collagen turnover [69]. IL-6 signals activate two major downstream pathways, the Janus kinase (JAK) signal transducer/ activator of transcription 3 (STAT3) pathway and the JAK-SH2 domain tyrosine phosphatase 2 (SHP2)-mitogen-activated protein (MAP) kinase pathway. Together with TGF-β, IL-6 drives differentiation of naive CD4-positive T cells into Th17 cells, which produce IL-17, an inflammatory cytokine [70]. Therefore, the utility of IL-6 as an SSc biomarker has been investigated. Skin samples from patients with early dcSSc revealed augmented IL-6 expression in fibroblasts, mononuclear cells, and endothelial cells [71]. Moreover, elevated serum IL-6 was found to correlate with the extent of skin involvement [71,72], SSc-ILD, and to portend poor long-term outcomes in SSc [71]. Elevated serum IL-6 levels in early dcSSc patients were also associated with more severe skin involvement and poor prognosis at the 3 year follow-up [71]. One study found that IL-6 serum levels >7.67 pg/mL correlated with increased mortality and marked lung function impairment. Among eight serum cytokines, chemokines, and growth factors (IL-6, IL-8, IL-10, CCL2, CXCL10, CX3CL1, FGF-2, and VEGF), only IL-6 was found to be an independent predictor of the DLco decline in both SSc-ILD and idiopathic pulmonary fibrosis [73]. A phase 3 trial of tocilizumab (anti-IL-6 receptor antibody) for SSc showed tocilizumab might preserve lung function in patients with early SSc-ILD [74].

### 5.2. B-Cell-Activating Factor Belonging to the Tumor Necrosis Factor Family (BAFF, also Known as BLyS) and a Proliferation-Inducing Ligand (APRIL)

The BAFF and APRIL cytokines are produced by various cells including monocytes and dendritic cells and both bind to each receptor expressed on B cells, known as B cell maturation protein (BCMA) and transmembrane activator and CAML interactor (TACI). However, the BAFF receptor 3 (BR3) expressed on B cells recognizes only BAFF. BAFF and APRIL have similar critical functions in B cell development and survival, Ig class switch, and costimulation. Augmented BAFF signaling has been implicated in the induction of B cell functional abnormalities, which indicates the potential for it to play a role in the development of SSc [75]. Furthermore, a recent study in the bleomycin-induced scleroderma mouse model demonstrated that inhibition of BAFF attenuates skin and lung fibrosis with reduction of IL-6–producing effector B cells [75]. 

Of note, serum levels of BAFF and APRIL were found to be elevated in patients with SSc. Furthermore, serum BAFF levels serve as a marker of severe skin sclerosis, whereas APRIL levels serve as a marker of pulmonary fibrosis [76,77].

## 6. Chemokines

Chemokines are more easily detected in peripheral blood than are cytokines or growth factors. Thus, several chemokines have been investigated as possible biomarkers of SSc.

### 6.1. CCL2 (Monocyte Chemoattractant Protein-1; MCP-1)

Mainly produced by macrophages, fibroblasts, endothelial cells and type II pneumocytes [78,79], the CCL2 chemokine plays a crucial role in leukocyte trafficking and activates monocytes and T cells resulting in type 2 polarization [80]. Additionally, CCL2 stimulates fibroblasts to differentiate towards myofibroblasts and to produce collagen via specific receptors and through endogenous upregulation of TGF-β expression [81].

In sclerotic skin of SSc patients, CCL2 expression was found to be augmented in the epidermis, inflammatory mononuclear cells, and vascular endothelial cells [82]. Further, serum levels of CCL2 were elevated in SSc patients and significantly correlated with the presence of ILD [82]. In other reports, elevated serum CCL2 levels were found to be associated with the severity of pulmonary injury and shorter survival in patients with SSc [78,83]. Furthermore, CCL2 concentrations in BAL fluids were associated with ILD and resulting computed tomography scores [84], and correlated negatively with lung function.

### 6.2. CXCL4 (Platelet Factor 4; PF-4)

CXCL4 is released from various cell sources, and particularly from aggregated platelets, and plays important roles in inflammation and wound repair through its chemotactic activity on neutrophils, monocytes, and fibroblasts [85]. Furthermore, this chemokine stimulates the expression of profibrotic cytokines including IL-4 and IL-13, and inhibits the expression of the antifibrotic IFN–γ. Moreover, CXCL4 can bind to glycosaminoglycans, such as chondroitin sulfate proteoglycan on neutrophils [86] and CXCR3A/B [87]. Both of which are considered to play a role in tissue-dependent fibrosis of SSc [88].

A proteome-wide study showed that SSc-derived plasmacytoid dendritic cells predominantly secrete CXCL4 [85]. Further, CXCL4 levels were markedly elevated in plasma of patients with SSc and highly correlated with skin fibrosis, ILD, and PAH [85], suggesting the utility of this marker to predict the progression of the lung involvement in SSc. More specifically, elevated plasma levels of CXCL4 were associated with a more rapid decline in DLco, whereas decreased levels of this chemokine were highly predictive of improved pulmonary function during immunosuppressive therapy [85].

### 6.3. CXCL10 (IFN-γ-Inducible Protein 10; IP-10)

CXCL10 is an IFN-inducible chemokine and potent chemoattractant for Th1 cells. Previously it was found that the plasma levels of chemokines induced by IFN correlates with the IFN gene expression signature (CXCL10 and CXCL11) and may be a promising biomarker of SSc [89]. Moreover, the IFN-inducible chemokine score was found to correlate with the Medsger severity index, particularly with regard to the involvement of skin, lung, and muscle.

A retrospective study showed that serum CXCL10 levels are increased in preclinical (nonfibrotic)/early SSc patients and that high concentrations of CXCL10 indicate a faster rate of progression from preclinical/early SSc to worse disease stages [90]. Another study demonstrated that higher serum CXCL10 values are found in SSc patients with lung and renal involvement [91,92]. On the other hand, two independent studies found no correlation between serum CXCL10 and the severity of lung involvement [73,79]. This discrepancy may result from type 1 cytokines playing a more important role during early disease stages and a lesser role in late-stages [91]. Nonetheless, establishment of the utility of CXCL10 as a biomarker will require a prospective longitudinal study.

### 6.4. CX3CL1 (Fractalkine)

CX3CL1 is a unique chemokine with dual functions in cell adhesion and migration [93]. Expressed on the surface of a vast majority of cell types, including endothelial cells, epithelial cells, macrophages and vascular smooth muscle cells, CX3CL1 interacts with its unique receptor, CX3CR1, expressed on monocytes/macrophages, NK cells, and a subpopulation of T cells. SSc patients with diffuse skin sclerosis, ILD, or digital ulcers were found to have increased serum CX3CL1 [94]. An association between elevated serum CX3CL1 and disease severity, increased frequency of pulmonary fibrosis, and lower DLco in lungs was also reported [95]. Furthermore, multivariable regression analysis suggested that serum CX3CL1 levels are associated with progression of ILD, but not with PAH [96]. Interestingly, serum CX3CL1 levels positively correlate with the increase in non-hematopoietic CD34^+^CD45^-^ endothelial progenitor cells, possibly reflecting vascular activation in SSc [95]. Moreover, augmented expression of CX3CL1 and an increased presence of CX3CR1^+^ leukocytes is seen in fibrotic skin and lung lesions of SSc patients. Yet another study demonstrated increased CX3CL1 concentrations in the lung tissues from SSc patients [96]. Further, a neutralizing anti-CX3CL1 monoclonal antibody was recently shown to protect against tissue fibrosis and vascular injury in a mouse model of skin fibrosis [97]. Therefore, CX3CL1 shows potential as both a therapeutic target and as a biomarker in SSc.

### 6.5. Combined Studies of Chemokines

A prospective study found that early SSc patients who had dcSSc and/or ILD showed increased serum CCL2, CCL5, CXCL8, CXCL9, and CXCL10 compared with healthy controls [98]. Of these, initial serum CXCL8 levels were the independent factor significantly associated with the health assessment questionnaire disability index (HAQ-DI) at the four year follow-up [98]. Thus, serum CXCL8 could be a prognostic indicator of physical dysfunction in SSc.

## 7. Adhesion Molecules

Increased numbers and activation of leukocytes are found in the circulation and in tissues of SSc patients [99,100]. Endothelial cell injury results in the induction of multiple adhesion molecules and subsequently, additional endothelial damage and subsequent tissue fibrosis occur due to the recruitment of inflammatory cells [101]. Therefore, circulating levels of various adhesion molecules may reflect the disease activity or severity of vascular injury and tissue fibrosis in SSc.

### 7.1. ICAM-1 (Intercellular Adhesion Molecule-1)

Intercellular adhesion molecule-1 (ICAM-1, CD54) is a cell surface glycoprotein that mediates strong adhesion of leukocytes to the endothelium and their trans-endothelial migration into sites of inflammation [102,103].

Due to increased surface expression of ICAM-1 SSc fibroblasts may possess an augmented potential to bind inflammatory cells [104]. SSc patients with diffuse rapidly progressive disease or digital ulcers were found to have increased soluble ICAM-1 in serum compared to that in healthy subjects [105,106].

### 7.2. Combined Studies of Adhesion Molecules

In a previous study, serum levels of ICAM-1, P-selectin, vascular cell adhesion molecule-1 (VCAM-1), and to a lesser degree, E-selectin were found to be well correlated with their in situ expression and with clinical disease activity in SSc [107]. SSc patients with renal crisis displayed elevated mean serum levels of E-selectin, ICAM-1, and VCAM-1. On the other hand, serum levels of these molecules were unchanged in lcSSc patients with PAH [108]. SSc patients with systemic organ involvement had elevated serum E-selectin, VCAM-1, VEGF and endothelin-1 compared to those without [109]. A small cohort study of SSc patients revealed increased serum levels of ICAM-1, platelet endothelial cell adhesion molecule-1, P-selectin, and VCAM-1 compared with healthy controls at baseline, all of which returned to normal levels following 12 months of treatment with bosentan, a dual antagonist of endothelin-1 for the endothelin-1 type A and type B receptors [110]. Serum levels of E-selectin, ICAM-1, and VCAM-1 were initially elevated in SSc, an effect that was reversed after iloprost (prostacyclin analogue) infusion to treat Raynaud’s phenomenon [111]. A multicenter, prospective, observational study revealed elevated serum levels of ICAM-1, P-selectin, and E-selectin and a reduction of serum L-selectin in patients with early SSc [112]. Further, baseline levels of serum ICAM-1 were inversely associated with subsequent pulmonary dysfunction in patients with early SSc, while P-selectin levels were significantly associated with physical disability (HAQ-DI) [112].

## 8. Vascular Biomarkers and Biomarkers of Endothelial Activation

One of the earliest clinical features of SSc to appear is vascular injury [113,114]. Microangiopathy is the marked decrease and chaotic architecture of capillaries and small vessels which leads to chronic tissue hypoxia. Despite this tissue hypoxia, compensative angiogenesis in SSc is insufficient likely due to an imbalance between angiogenic and angiostatic factors [115]. In addition, endothelial damage during SSc causes vascular fibroproliferative lesions in multiple organs, resulting in outcomes including PAH and renal crisis [116]. Moreover, through a process known as endothelial mesenchymal transition endothelial cells or pericytes contribute to the development of tissue fibrosis in SSc [117]. Numerous potential biomarkers for the endothelial cell injury characteristic of SSc have been proposed and will be discussed here [117].

### 8.1. Endostatin

The carboxyl-terminal fragment of type XVIII collagen is known as endostatin, which inhibits angiogenesis by blocking the activity of VEGF and bFGF [118]. While not significantly elevated in SSc, serum endostatin is linked with the occurrence of giant capillaries as assessed by nailfold capillaroscopy [119]. Plasma levels of endostatin were markedly elevated in patients with SSc and correlated positively with right ventricular systolic pressure [120].

### 8.2. Endoglin

Part of the TGF-β receptor complex on endothelial cells, endoglin is a transmembrane glycoprotein with a critical role in angiogenesis and tissue remodeling [121]. Multivariate analysis of a large SSc cohort revealed significantly increased soluble endoglin in SSc patients with digital skin ulcers, anticentromere antibodies, and reduced DLco divided by alveolar volume [122]. Further, the soluble endoglin concentration in serum was significantly increased in lcSSc patients compared with those with dcSSc, systemic lupus erythematosus, or normal controls [123]. Patients with elevated soluble endoglin levels more frequently had telangiectasia than did those with normal soluble endoglin levels. In patients with lcSSc, soluble endoglin levels positively correlated with pulmonary artery pressure. In addition, mutations in the endoglin gene cause a form of hereditary hemorrhagic telangiectasia (Osler–Weber–Rendu syndrome) and a Mendelian autosomal vascular disorder [124]. In addition, it has been reported that a polymorphism of the endoglin gene is associated with SSc-related PAH [125].

### 8.3. vWF (Von Willebrand Factor)

Platelet adhesion to injured vessel walls is mediated by vWF, a multimeric plasma glycoprotein synthesized by endothelial cells, which serves as a carrier and stabilizer for coagulation factor VIII. Patients with SSc or Raynaud’s phenomenon exhibit increased vWF in plasma [126,127,128,129], an effect associated with disease severity [126], pulmonary involvement [127], and extent of ILD as determined by radiography [128]. Moreover, lcSSc patients at risk for future development of elevated pulmonary arterial pressure can be identified by serum vWF levels [130]. In addition, plasma levels of ADAMTS-13 (a disintegrin and metalloproteinase with a thrombospondin type 1 motif, member 13), which specifically cleaves vWF, were significantly decreased in patients with SSc [131].

## 9. Biomarkers of SSc-ILD

Due to the unreliability of radiographic evaluation via computed tomography (CT) scan for diagnosis of ILD, the main cause of SSc-related death, there is an urgent need for serum biomarkers for this condition. Proteins secreted by alveolar epithelial cells, inflammatory cytokines, and chemokines have been the main focus of the search for serum biomarkers of ILD.

### 9.1. Krebs von den Lungen-6 (KL-6)

Mainly produced by alveolar type II pneumocytes and respiratory bronchiolar epithelial cells, KL-6 is a high molecular weight glycoprotein normally involved in fibroblast stimulation and inhibition of apoptosis [132,133]. Currently, KL-6 is considered to be one of the most reliable serum marker for ILD as several studies have confirmed elevated serum KL-6 in patients with ILD, including SSc-ILD [132,134,135]. Two small cohort studies determined SSc-ILD could be diagnosed with 93% accuracy with sensitivity and specificity of 79 and 93%, respectively, based on a KL-6 serum value of 500 U/mL [136,137]. However, this was not supported by a recent large prospective study that reported a sensitivity of only 44% and a specificity 85% for a serum cut-off value of 923 U/mL [138]. Nonetheless, serum levels of KL-6 may also indicate disease severity, as this glycoprotein is negatively correlated with pulmonary function and correlates positively with radiological evidence of impairment or the presence of extensive lung fibrosis in SSc-ILD [56,138]. Further, in patients suffering from ILD, a KL-6 serum level >1000 U/mL was associated with increased mortality at 5 years of follow-up [139]. Similarly, a serum level >1273 U/mL was found in SSc patients with end-stage SSc-ILD [140]. In several studies, the fluctuation of KL-6 serum levels was found to be associated with flare-ups or disease improvements. Furthermore, a small retrospective study suggested a KL-6 serum level >2000 U/mL in SSc patients under treatment (corticoids and cyclophosphamide) was indicative of poor therapeutic response [141]. On the other hand, in a recent prospective large cohort study no correlation was found between KL-6 and mortality or with overall prognosis [138]. Thus, KL-6 could be a reliable biomarker to assess SSc-ILD severity; however, further studies are necessary to confirm its utility for diagnosis, prognosis, and prediction of therapeutic responsiveness.

### 9.2. Surfactant Protein-A and D (SP-A, SP-D)

SP-A and SP-D are pulmonary surfactant lipoproteins secreted by alveolar type II pneumocytes and Clara cells [142] that function to reduce alveolar surface tension at the air/liquid interface thereby precluding small airway collapse. The serum values for these pneumo-specific proteins were found to reflect the extent of damage to the capillary/alveolar barrier [143]. SP-D levels were found to be elevated in sera from patients with ILD as well as SSc-related ILD [132,135,142] and thus, appear to reflect the severity of ILD in SSc [135,144,145]. Comparison of SP-D with KL-6 as biomarkers in SSc revealed SP-D to be more sensitive, but less specific, for ILD than KL-6 [144]. However, the diagnostic and prognostic value of SP-D as a biomarker in SSc-ILD is somewhat controversial [129,144,145,146,147,148]. Recently, a large prospective cohort study (*n* = 427) showed increased serum SP-D values in concert with anti-topoisomerases I antibody were detectable in SSc-ILD with 97% sensitivity and 69% specificity [138]. Thus, this allowed prediction of the occurrence of ILD with an accuracy of 80% while ILD could be ruled out with an accuracy of 95%. Nevertheless, SP-D was not correlated with the severity or progression of lung impairment leading to mortality in individual patients [138]. Following cyclophosphamide and prednisolone treatment a rapid decrease in serum SP-D (below 200 ng/mL) was found to be predictive of a positive treatment response [141]. Large scale prospective studies are necessary to determine the efficacy of SP-D as a biomarker for evaluating and predicting therapeutic responses. With respect to SP-A, which is known to also be associated with alveolar epithelial damage, for prediction of SSc-ILD, its sensitivity and specificity is lower than that of SP-D [141]. A cohort study conducted by the Scleroderma Lung Study Research Group also suggested the fundamental necessity of highly sensitivity and specificity of both KL-6 and SP-D as a combined predictive marker of alveolitis in SSc-ILD. In this study KL-6 and SP-D levels were found to correlate significantly with maximum fibrosis scores, but not with maximum ground-glass opacities, as revealed by high-resolution CT [138]. Therefore, KL-6 and SP-D may not necessarily reflect the radiographic activity of ILD.

### 9.3. CCL18 (Pulmonary and Activation Regulated Chemokine (PARC))

Another promising biomarker for SSc-ILD is CCL18. Chiefly produced by macrophages and dendritic cells in the lungs, CCL18 is a constitutively expressed chemokine that selectively drives T cell chemotaxis [149]. High concentrations of CCL18 induce collagen production in primary pulmonary fibroblasts [150]. Even after correction for baseline ILD severity, CCL18 was found to be a suitable prognostic marker in SSc-ILD [138,151]. Moreover, CCL18 is involved in various lung fibrosis disorders in addition to SSc-ILD including hypersensitivity pneumonia, sarcoidosis, and idiopathic lung fibrosis [152]. Further, dermal fibroblasts from SSc skin proliferate and produce collagen in response to CCL18, and through positive feedback native collagen increases CCL18 production [151].

Marked elevation of serum CCL18 occurs in association with the development of ILD in a manner closely correlated with ILD severity [153,154,155]. In addition, in SSc patients the serum CCL18 concentration is a meaningful predictor of mortality and progression of ILD [154,156], and the level of CCL18 produced by BAL cells and in serum reflects the fibrotic activity seen in SSc-ILD [151]. SP-D reportedly correlates with concurrently obtained FVC, and CCL18 is a prognosticator of short-term decline in FVC [146]. However, in this study the long-term course of FVC could not be predicted by either SP-D nor CCL18 in early SSc patients. A large prospective trial (*n* = 427) confirmed these results by showing that a high CCL18 serum baseline value (cut-off 84 pg/mL) (HR = 2.9) and male sex (HR = 2.48) were the best predictors of subsequent lung function decline [138], although the association with SSc-relevant mortality was unclear. Taken together, these results suggest that CCL18 can be a gauge of the activity or severity of SSc-ILD.

## 10. Biomarkers of Pulmonary Arterial Hypertension

PAH is one of the most important factors affecting morbidity and mortality in SSc. Because PAH is asymptomatic in the early stages its diagnosis is often delayed. In addition, there is a lack of validated laboratory examinations or serologic biomarkers sensitive to and specific for the diagnosis of PAH. Therefore, biomarkers for early PAH are coveted.

### 10.1. Brain Natriuretic Peptide (BNP) and N-Terminal-Pro Hormone BNP (NT-proBNP)

Ventricular myocytes secrete BNP and NT-proBNP in a manner indicative of myocardial responses to stretch, hypoxia, and certain neurohormonal stimuli [157]. SSc patients with an NT-proBNP in excess of 395 pg/mL are highly likely to have pulmonary hypertension (sensitivity 56%, specificity 95%). Furthermore, survival can be predicated by baseline and serial changes of NT-proBNP levels [158]. Because BNP and NT-proBNP serum levels are inclined to increase in SSc patients with early PAH and correlate with estimated pulmonary arterial pressure they are considered valuable biomarkers for PAH [159,160]. Moreover, decreased DLco/alveolar volume ratio and an increased NT-proBNP are predictors of PAH in SSc [161], Treatment guidelines of the Task Force for the Diagnosis of Treatment of PAH of the European Society of Cardiology and European Respiratory Society indicate that plasma levels of BNP and NT-proBNP are key parameters for assessing the severity, stability, and prognosis of PAH [162]. However, plasma levels of BNP or NT-proBNP are elevated in various heart diseases including congestive heart failure and not specific for PAH.

### 10.2. Endothelin-1

Cells including activated endothelial cells, macrophages, and fibroblasts produce the vasoconstrictor peptide Endothelin-1. Expression of endothelin-1 is induced by TGF-β, and its signaling via the endothelin receptors A and B results in fibroblast migration, myofibroblast differentiation, and proliferation of smooth muscle cells [163], while its induction of vasoconstriction mainly occurs via the endothelin receptor type A. Therefore, in SSc it is established that endothelin-1 plays a critical role in proliferative vasculopathies such as PAH [164]. In addition, blockade of the endothelin-1-receptor is highly effective for treatment of PAH.

Patients with SSc reportedly display increased plasma levels of endothelin-1 [165,166,167], which positively correlates with systolic pulmonary arterial pressure [167]. In particular, SSc patients with advanced microangiopathy express elevated plasma endothelin-1 as defined by capillaroscopy. In addition, a positive linear correlation between endothelin-1 levels and systolic pulmonary arterial pressure was found [167,168,169]. Endothelin-1 is increased both in lung tissues [170] and BAL fluid [171] of patients with SSc-ILD. However, serum endothelin-1 concentration was not found to correlate with severity of SSc-ILD [171,172]. Furthermore, treatment with bosentan, a dual endothelin-1 receptor antagonist, did result in improvement of SSc-ILD [173,174]. Thus, the level of circulating endothelin-1 may be useful for evaluating SSc-related PAH. Additionally, when the baseline occurrence of active digital ulcers or previous history of digital ulcers is taken into account, anti-endothelin 1 type A receptor autoantibody is an independent predictor of the newly developed ischemic digital ulcers [175].

## 11. Collagen

### 11.1. Type I Collagen Degradation

Tissue fibrosis due to deposition of extracellular matrix protein (ECM) is the typical pathology seen with SSc. The most plentiful ECM protein deposited in fibrotic skin of SSc is type I collagen. In fact, the excessive synthesis and deposition of collagens were the rationale for conducting turnover measurements to assess the activity and severity of SSc.

Serum concentrations of the C-terminal telopeptide of type I collagen, a marker of type I collagen degradation, were found to be closely associated with the extent of skin fibrosis in SSc patients [176]. Another study showed that serum concentrations of this telopeptide correlate significantly with the modified Rodnan total skin thickness score (mRSS) and acute phase reactants in patients with SSc [177].

### 11.2. Type III Collagen Degradation

Serum levels of the N-terminal type III procollagen peptide were found to be reflective of disease activity in SSc [178]. In addition, the N-terminal type III procollagen peptide was found to correlate with high resolution computed tomography scores in cases of SSc with pulmonary involvement [179]. In a prospective follow-up study, increased levels of the N-terminal propeptide of type III procollagen was an independent, unfavorable prognostic sign [179]. Levels of type III and VI collagens, but not type IV collagen, were found to correlate with mRSS and were elevated in dcSSc patients. In addition, a constant decrease of serum type III collagen was noted in SSc over the clinical course [180].

## 12. Matrix Metalloproteinases (MMPs)

MMPs are proteolytic enzymes secreted by macrophages, fibroblasts, and endothelial cells that play important roles in turnover of the extracellular matrix [181,182]. In SSc-ILD, chronic microinjuries to the lung result in architecturally disrupted and collagen-rich ECM caused by the interaction of cells in the epithelial, endothelial, and interstitial compartments with components of the innate and adaptive immune systems. Although MMPs were first described as proteases that degrade ECM, these enzymes play multifunctional roles including inducing the release and activation of cytokines and growth factors [183]. Moreover, their biochemical activities are kept in check by protease inhibitors (tissue inhibitors of metalloproteinases) in order to maintain homeostasis during remodeling of the extracellular matrix.

### 12.1. MMP-7

Serum levels of MMP-7 were found to be higher in SSc patients than in healthy subjects, with much higher concentrations in patients with SSc with ILD [184,185,186]. Other studies identified a correlation between MMP-7 levels and disease severity [185,186]. Combined measurements of KL-6 and MMP7 have been suggested as a useful monitoring tool for identifying SSc patients at high risk of developing clinically significant ILD [147].

### 12.2. MMP-9

Substrates of MMP-9 include type IV collagen in the basement membrane, and this enzyme has been implicated in the pathogenesis of cancer, autoimmune diseases, and various pathologic conditions characterized by excessive fibrosis. Dermal fibroblasts from SSc patients overproduce MMP-9 when stimulated with IL-1β, tumor necrosis factor-α, or TGF-β [184]. Further, SSc patients were found to have higher concentrations of serum MMP-9, which was significantly higher in dcSSc compared with lcSSc [184]. In addition, serum levels of MMP-9 correlate well with the extent of skin sclerosis as determined by mRSS [184].

### 12.3. MMP-12

The expression of MMP-12 is augmented in SSc skin fibroblasts and endothelial cells [187]. This enzyme degrades various ECM components including type IV collagen, elastin, and fibronectin [188]. Serum MMP-12 levels were significantly increased in dcSSc patients, and were associated with skin sclerosis, presence of digital ulcers, severity of lung restriction, and nailfold bleeding [186]. MMP-12 can be found at greater concentrations in serum of SSc patients with ILD than in that of SSc patients without ILD. Further, there is a close association between lower FVC in patients with SSc-ILD and MMP-12 levels [186].

## 13. MicroRNAs (miRNAs)

miRNAs are non-coding RNAs 18–23 nucleotides in length that function as intracellular regulators of gene expression. Their main role is in the downregulation of protein translation by destabilizing mRNA and thus are post-transcriptional regulators of gene expression [189]. For example, miR-21 suppresses baseline expression of the anti-fibrotic signaling molecule SMAD7, thereby promoting expression of profibrotic genes. Consistent with this, expression of miR-21 is elevated in fibroblasts from SSc skin. By contrast, miR-29 has inhibitory effects on fibrotic gene expression and its expression is reduced in fibroblasts isolated from SSc patients [190,191]. Increased expression of miR-155 in lung tissue is associated with impaired respiratory function and increased lung fibrosis [192].

Although miRNA expression is tissue-specific and cell-type-dependent, the circulating fraction of miRNAs may act as biomarkers [193]. For example, serum levels of miR-29a were found to be reduced in patients with SSc and are assumed to be relevant as an actor for developing SSc in the early disease stage [194]. SSc patients exhibit significantly lower serum levels of miR-92-a and miR-142-3p compared to healthy subjects, an effect specific to SSc compared to other autoimmune diseases, including dermatomyositis and systemic lupus erythematosus. Although much remains to be learned about the spectrum of the aberrant regulation of miRNAs and the mechanisms of their action in SSc, these regulatory non-coding RNAs might prove useful as biomarkers and therapeutic targets.

## 14. Other Molecules

### 14.1. C-Reactive Protein (CRP)

CRP is an acute-phase protein with increasing serum concentration during inflammation, which is thought to contribute to host defense and other adaptive capabilities [195]. With respect to SSc, serum levels of CRP are associated with the progression of multi-organ involvement including skin sclerosis, PAH, and renal dysfunction [196,197]. In addition, elevated plasma CRP is associated with the risk of progressive early SSc-ILD [34]. SSc patients with high serum CRP levels (>8 mg/L) developed ILD more frequently and with worse pulmonary function and higher mortality than other patients (CRP < 8 mg/L) [198]. Furthermore, a small retrospective study revealed that elevated baseline levels of CRP are predictive of a poor therapeutic response [199].

### 14.2. Soluble CD163

Macrophages can be functionally distinguished into two subsets, M1 and M2 macrophages. M1 macrophages promote inflammation through secretion of high levels of proinflammatory cytokines. By contrast, M2 macrophages exhibit an anti-inflammatory function and contribute to wound healing. Moreover, excessive infiltration of M2 macrophages can induce fibrosis of various tissues and may play a crucial role in SSc pathogenesis [200,201].

CD163, a hemoglobin scavenger receptor, is a representative marker expressed on activated M2 macrophages [202,203]. In response to oxidative stress or inflammatory stimuli a soluble form of CD163 (sCD163) is released from the cell surface of M2 macrophages by proteolysis [204,205]. This soluble sCD163 is significantly elevated in the sera and skin of SSc patients compared with normal controls [206,207,208]. Previous studies reported an association between sCD163 in serum and the severity of PAH in SSc [207,209]. Further, serum sCD163 concentrations in SSc patients with ILD are significantly elevated relative to those in SSc patients without ILD [210]. While elevated serum sCD163 levels were associated with severe skin sclerosis, this was also indicative of a lower risk of digital ulceration [211]. These findings support sCD163 as a potential biomarker in SSc. However, it should be noted that another study could not find any relationship between sCD163 concentrations and lung vascular disease, digital ulcers, or any clinical or laboratory variables [208]. Thus, more studies are needed to determine its utility as a biomarker.

### 14.3. YKL-40 (Chitinase-3–Like Protein 1)

YKL-40 belongs to the family of mammalian chitinase-like proteins, regulates cell proliferation and survival, and is produced by activated macrophages [212]. A growth factor for connective tissue [213], YKL-40 has been shown to exert a promitogenic effect on lung fibroblasts in SSc animal models [214,215]. Several studies reported that serum YKL-40 levels are elevated in patients with SSc compared with healthy subjects [216,217]. Moreover, a prospective study showed that serum YKL-40 levels were higher in SSc patients with pulmonary involvement than in SSc patients without it [218]. In addition, elevated serum YKL-40 values in SSc are associated with worse pulmonary involvement and a higher mortality rate [218].

## 15. Limitations

One of the major limitations of this review is that candidate biomarkers of SSc were chosen according to our opinions with regard to their potential usefulness. Another limitation is the lack of longitudinal studies for some biomarkers, which reduces the clinical impact of the findings.

## 16. Conclusions

Findings from various perspectives have increased the utility of potential SSc biomarkers as diagnostic and prognostic tools. Large, multicenter, prospective studies of well-defined clinical cohorts should be performed to fully establish useful and reliable biomarkers of SSc.

## Figures and Tables

**Table 1 jcm-09-03388-t001:** Potential serum/plasma biomarkers of systemic sclerosis.

Biomarker	Clinical Association
TGF-β↑	Digital ulcers, dcSSc
TGF-β↓	dcSSc, mRSS (in dcSSc)
VEGF↑	Systemic organ involvement, PAH, shorter disease duration, skin sclerosis, reduced capillary density of nailfold
VEGF↓	Digital ulcers
CTGF↑	mRSS, ILD
GDF-15↑	Skin sclerosis, PAH, ILD, respiratory dysfunction (FVC, DLco)
IL-6↑	mRSS, early progressive skin sclerosis, poor prognosis, DLco decline in SSc-ILD
BAFF↑	Skin sclerosis
APRIL↑	Pulmonary fibrosis
CCL2↑	ILD (lung dysfunction, CT scores), mRSS
CXCL4↑	mRSS, lung fibrosis, PAH, disease progression
CXCL8↑	Predictive of physical dysfunction
CXCL10↑	Preclinical/early SSc
CX3CL1↑	dcSSc, ILD, digital ulcer
ICAM-1↑	Rapidly progressive disease, digital ulcers, dcSSc, ILD, joint involvement, renal crisis, predictive of respiratory dysfunction
VCAM-1↑	Systemic organ involvement, renal crisis, disease activity
E-selectin↑	Systemic organ involvement, renal crisis, disease activity
P-selectin↑	Disease activity, predictive of physical disability
endostatin↑	PAH
endoglin↑	lcSSc, anticentromere Ab, cutaneous ulcer, telangiectasia, PAH.
Von Willebrand factor↑	Raynaud’s phenomenon, disease severity, ILD, predictive of PAH
KL-6↑	Severity of ILD, maximum fibrosis scores on HRCT
SP-D↑	Severity of ILD, maximum fibrosis scores on HRCT
CCL18↑	Activity and severity of ILD, predictive worsening of ILD and mortality
BNP/NT pro-BNP↑	Severity, stability, and prognosis of PAH
Endothelin-1↑	PAH, systemic organ involvement, microangiopathy defined by capillaroscopy
Type I collagen (C-terminal telopeptide)↑	Skin fibrosis, mRSS, pulmonary dysfunction, CRP
Type III collagen (N-terminal peptide)↑	Disease activity, mRSS, HRCT score, prognosis
MMP-7↑	ILD, disease severity
MMP-9↑	mRSS, dcSSc
MMP-12↑	Skin sclerosis, dcSSc, ILD, nailfold bleeding, lower FVC
CRP↑	Skin sclerosis, PAH, renal dysfunction, risk of progressive early ILD, worse pulmonary function
sCD163↑	ILD, PAH, skin sclerosis
YKL-40↑	Pulmonary involvement, higher mortality rate

↑, upregulated; ↓, downregulated; TGF-β, transforming growth factor; GDF-15, growth differentiation factor 15; BAFF, B-cell-activating factor belonging to the tumor necrosis factor family; APRIL, a proliferation-inducing ligand; MMP, matrix metalloproteinases; BNP, brain natriuretic peptide; NT-proBNP, N-terminal-pro hormone BNP; CTGF, connective tissue growth factor; mRSS, modified Rodnan total skin thickness score; ILD, interstitial lung disease; IL-6, interleukin 6; DLco, diffusing capacity of carbon monoxide; CT, computed tomography; PAH, pulmonary arterial hypertension; ICAM-1, intercellular adhesion molecule 1; dcSSc, diffuse cutaneous systemic sclerosis; VEGF, vascular endothelial growth factor; lcSSc, limited cutaneous systemic sclerosis; KL-6, krebs von den Lungen-6; HRCT, high resolution CT; SP-D, surfactant protein-D.

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
