# Peer review of "Potential Biomarkers in Systemic Sclerosis: A Literature Review and Update"

_jcm, 2020, doi:10.3390/jcm9113388_

Round 1

Reviewer 1 Report

Globally, I think that the manuscript is too much long, especially the first part, divided by type of biomarkers. It is very confusing and not helpful for the reader. I would have saved only the table to summarize this part. Instead I would have maintained only the part regarding biomarkers divided for single organ involvement, as it could be more helpful for the reader.

Moreover, even if the purpose is not that of a systematic literature review, the methods and the objective for this literature review are even not described at all.

Some other minor comments:

Please provide a reference for this sentence “The clinical course of skin sclerosis and organ lesions varies from case to case. Currently, SSc related mortality is predominantly due to ILD and PAH.”

Since actually there is no demonstration that an early treatment could improve the outcome of Systemic Sclerosis patients, this sentence  “Therefore, it is essential to distinguish high risk patients in order to initiate therapeutic interventions prior to disease advancement” should be substantially modified. In fact, the importance of biomarkers in the context to SSc is to identify predictors of the evolution of the disease, for example to predict the onset of ILD in one patient and more importantly to predict the evolution of ILD. This part should be enriched.

Instead of “Similar to what is seen with other autoimmune diseases” (line 50), I would say “Similarly to what is seen in other connective tissue diseases”.

Line 53: I would have added that these 3 auto-antibodies are specific and included in 2013 ACR/EULAR criteria.

Anti-Topo I are associated with diffuse cutaneous involvement.

Anti-RNA Polymerase are associated with malignancies, but only with malignancies synchronous to scleroderma onset, that is currently their main key role as biomarkers. If such positivity is found, clinicians should seek for the presence of an occult cancer (if not already manifested) and monitor for the onset of SRC.

Line 77: a reference is missing

When talking about Il-6 I would have cited the very recent results from the phase 3 trial on anti IL-6 Tocilizumab, as previously mentioned for PDGF and Nintedanib.

Author Response

Response to Reviewer 1 Comments

Globally, I think that the manuscript is too much long, especially the first part, divided by type of biomarkers. It is very confusing and not helpful for the reader. I would have saved only the table to summarize this part. Instead I would have maintained only the part regarding biomarkers divided for single organ involvement, as it could be more helpful for the reader. Moreover, even if the purpose is not that of a systematic literature review, the methods and the objective for this literature review are even not described at all.

We are extremely grateful for crucial suggestions of the reviewer. However, there are many biomarkers that overlap functionally in distinct organs and are not easy to classify for each organ. For this reason, we would like to maintain the current structure. As the reviewer pointed out, we have added the methods and the objective in the text as follows (page 2, line 5-10).

In this article, we aimed to literature review major candidate biomarkers in SSc according to their distinct biological actions. A PubMed search for articles published between January 1981 and August 2020 was conducted using the following keywords: “systemic sclerosis” and “biomarker”. From those references, a possible broad spectrum of biomarkers were selected subjectively. Some reference lists of identified articles were searched for further articles.

Point 1: Please provide a reference for this sentence “The clinical course of skin sclerosis and organ lesions varies from case to case. Currently, SSc related mortality is predominantly due to ILD and PAH.”

Response 1: We appreciate for the reviewer’s suggestion. We have added the reference (page 1, line 39).

Point 2: Since actually there is no demonstration that an early treatment could improve the outcome of Systemic Sclerosis patients, this sentence “Therefore, it is essential to distinguish high risk patients in order to initiate therapeutic interventions prior to disease advancement” should be substantially modified. In fact, the importance of biomarkers in the context to SSc is to identify predictors of the evolution of the disease, for example to predict the onset of ILD in one patient and more importantly to predict the evolution of ILD. This part should be enriched.

Response 2: We appreciate for the reviewer’s excellent suggestion. We have modified the sentence (page 1, line 44- page 2, line3).

Point 3: Instead of “Similar to what is seen with other autoimmune diseases” (line 50), I would say “Similarly to what is seen in other connective tissue diseases”.

Response 3: We have revised the sentences as the reviewer pointed out (page 2, line 13). Thank you very much.

Point 4: Line 53: I would have added that these 3 auto-antibodies are specific and included in 2013 ACR/EULAR criteria.

Anti-Topo I are associated with diffuse cutaneous involvement.

Anti-RNA Polymerase are associated with malignancies, but only with malignancies synchronous to scleroderma onset, that is currently their main key role as biomarkers. If such positivity is found, clinicians should seek for the presence of an occult cancer (if not already manifested) and monitor for the onset of SRC.

Response 4: We are grateful for your critical suggestion. We have added some sentences as pointed out (page 2, line 18- 41).

Point 5: Line 77: a reference is missing

Response 5: We appreciate for the reviewer’s suggestion. We have added the reference (page 2, line 44).

Point 6: When talking about Il-6 I would have cited the very recent results from the phase 3 trial on anti IL-6 Tocilizumab, as previously mentioned for PDGF and Nintedanib.

Response 6: We are very grateful to the reviewer’s important comments. We have added some comments regarding results from the phase 3 trial on Tocilizumab in the section (page 5, line 22-23).

Reviewer 2 Report

As general comments, the review is well structured and written, and meets the objectives of a review published in Journal of Clinical Medicine. Even if useful and reliable SSc biomarkers do not appear from here well-defined, because further studies are needed, the content of this review is clear and provides adequate information on potential SSc biomarkers discussing their clinical utility.

Minor comments:

1) Line 92: I suggest to entitle this paragraph just “Growth factors” since the following one analyze Cytokines and is indeed so entitled.

2) Paragraph “Connective tissue growth factor (CTGF)/CCN2" (Line 130): Since the role of CTGF is well explained, it should be appropriate to include in this section a work documenting further significant contributions to the concept described about CCN2. For example the following one: “Serratì S, Chillà A, Laurenzana A, Margheri F, Giannoni E, Magnelli L, Chiarugi P, Dotor J, Feijoo E, Bazzichi L, Bombardieri S, Kahaleh B, Fibbi G, Del Rosso M. Systemic sclerosis endothelial cells recruit and activate dermal fibroblasts by induction of a connective tissue growth factor (CCN2)/transforming growth factor β-dependent mesenchymal-to-mesenchymal transition. Arthritis Rheum. 2013 Jan;65(1):258-69.” This study indicates that CCN2 overproduced by SSc MVECs induced an efficient fibroblast activation, resulting also in an increase in fibroblast migration. CCN2 effects were mediated by an up‐regulation of the TGFβ system (ligand/receptors). Further, the CCN2/TGFβ‐dependent increase in fibroblast mobilization was accounted for by augmentation of their mesenchymal style of migration. Taken together, these data indicate that ECs of SSc patients may trigger fibrosis initiation by inducing a CCN2/TGFβ‐dependent fibroblast mesenchymal‐to‐mesenchymal transition, that is, an increase of their mesenchymal properties.

3) Line 188: Since it is the first time that DLco appears in the text, please specify the meaning here and not after in line 256.

Author Response

Response to Reviewer 2 Comments

As general comments, the review is well structured and written, and meets the objectives of a

review published in Journal of Clinical Medicine. Even if useful and reliable SSc biomarkers do

not appear from here well-defined, because further studies are needed, the content of this review is clear and provides adequate information on potential SSc biomarkers discussing their clinical utility.

We appreciate reviewer’s positive comments.

Point 1: Line 92: I suggest to entitle this paragraph just “Growth factors” since the following one analyze Cytokines and is indeed so entitled.

Response 1: Thank you very much for your critical suggestions. We have changed the entitle (page 3, line 12).

Point 2: Paragraph “Connective tissue growth factor (CTGF)/CCN2" (Line 130): Since the role of CTGF is well explained, it should be appropriate to include in this section a work documenting further significant contributions to the concept described about CCN2. For example the following one: “Serratì S, Chillà A, Laurenzana A, Margheri F, Giannoni E, Magnelli L, Chiarugi P, Dotor J, Feijoo E, Bazzichi L, Bombardieri S, Kahaleh B, Fibbi G, Del Rosso M. Systemic sclerosis endothelial cells recruit and activate dermal fibroblasts by induction of a connective tissue growth factor (CCN2)/transforming growth factor β-dependent mesenchymal-to-mesenchymal transition. Arthritis Rheum. 2013 Jan;65(1):258-69.” This study indicates that CCN2 overproduced by SSc MVECs induced an efficient fibroblast activation, resulting also in an increase in fibroblast migration. CCN2 effects were mediated by an up‐regulation of the TGFβ system (ligand/receptors). Further, the CCN2/TGFβ‐dependent increase in fibroblast mobilization was accounted for by augmentation of their mesenchymal style of migration. Taken together, these data indicate that ECs of SSc patients may trigger fibrosis initiation by inducing a CCN2/TGFβ‐ dependent fibroblast mesenchymal‐to‐mesenchymal transition, that is, an increase of their mesenchymal properties.

Response 2: We are extremely grateful for crucial suggestions of the reviewer and fully agree to the reviewer’s comments. We have revised this section (page 4, line 14-21).

Point 3: Line 188: Since it is the first time that DLco appears in the text, please specify the meaning here and not after in line 256.

Response 3: We corrected the sentences as the reviewer pointed out (page 4, line 42-43 and page 6, line 47).

Reviewer 3 Report

The work is a valuable contribution to modern knowledge of systemic sclerosis. Based on the latest literature, the authors presented a number of biomarkers of organ involvement in systemic sclerosis. 
Any such review is highly subjective, however, the authors highlighted this in the limitations section and selected a possible broad spectrum of biomarkers from antibodies to cytokines and vascular damage markers. A tabular summary is a great advantage of the work.
Minor comments on the introduction - requires linguistic refinement.

Author Response

Response to Reviewer 3 Comments

The work is a valuable contribution to modern knowledge of systemic sclerosis. Based on the latest literature, the authors presented a number of biomarkers of organ involvement in systemic sclerosis. Any such review is highly subjective, however, the authors highlighted this in the limitations section and selected from antibodies to cytokines and vascular damage markers.

We are grateful for positive comments of the reviewer.

Minor comments on the introduction - requires linguistic refinement.

Response: We have revised the introduction section (page 1, line 27-page 2, line 3). Thank you very much for your suggestion.

Round 2

Reviewer 1 Report

Thank you for submitting this revised version of the manuscript.

I have no other comments.

This manuscript is a resubmission of an earlier submission. The following is a list of the peer review reports and author responses from that submission.